# Effects of Land Use and Pollution Loadings on Ecotoxicological Assays and Bacterial Taxonomical Diversity in Constructed Wetlands

Subhomita Ghosh Roy [1,*], Charles F. Wimpee [1], Stephen A. McGuire [1,2] and Timothy J. Ehlinger [1,2]

[1] Department of Biological Sciences, University of Wisconsin-Milwaukee, P.O. Box 413, Milwaukee, WI 53201-0413, USA; cwimpee@uwm.edu (C.F.W.); smcguire@uwm.edu (S.A.M.); ehlinger@uwm.edu (T.J.E.)

[2] Institute for Systems Change and Peacebuilding, University of Wisconsin Milwaukee, P.O. Box 413, Milwaukee, WI 53201-0413, USA

\* Correspondence: ghoshroy@uwm.edu

**Abstract:** Freshwater ecosystems are affected by anthropogenic alterations. Different studies have extensively studied the concentrations of metals, nutrients, and water quality as measurements of pollution in freshwater ecosystems. However, few studies have been able to link these pollutants to bioindicators as a risk assessment tool. This study aimed to examine the potential of two bioindicators, plant ecotoxicological assays and sediment bacterial taxonomic diversity, in ecological risk assessment for six freshwater constructed wetlands in a rapidly urbanizing watershed with diverse land uses. Sediment samples were collected summer, 2015 and 2017, and late summer and early fall in 2016 to conduct plant ecotoxicological assays based on plant (*Lepidium*, *Sinapis* and *Sorghum*) growth inhibition and identify bacterial taxonomical diversity by the 16S rRNA gene sequences. Concentrations of metals such as lead (Pb) and mercury (Hg) (using XRF), and nutrients such as nitrate and phosphate (using HACH DR 2800™ spectrophotometer) were measured in sediment and water samples respectively. Analyses of response patterns revealed that plant and bacterial bioindicators were highly responsive to variation in the concentrations of these pollutants. Hence, this opens up the scope of using these bioindicators for ecological risk assessment in constructed freshwater wetland ecosystems within urbanizing watersheds.

**Keywords:** bioindicators; bacterial diversity; ecotoxicology; ecological risk assessment; wetlands; land use; pollutants

## 1. Introduction

Decades of industrialization, agriculture, and urbanization have resulted in toxic discharges such as metals, petroleum products, domestic wastes, nutrients, and other pollutants finding their way into freshwater ecosystems [1,2]. Studies [1,3,4] have shown that metals and nutrients may be indicators of pollutant levels in freshwater ecosystems. Various methods, including chemical indicators, have been used as "weight of evidence" to measure the extent of risk caused by anthropogenic pollutants and stressors [1]. What is lacking in the management of freshwater ecosystems is the extensive use of diverse bioindicators and linking them to pollutants as ecological risk assessment tools.

To explore this possibility, our study examined six freshwater constructed wetlands along the Pike River, Racine, WI, USA, in the southwestern portion of the Lake Michigan watershed (Figure 1). These wetland sites, originally built between 2001 and 2008, are connected to the river, and primarily serve as stormwater retention wetlands to reduce problems related to flooding in the surrounding watershed [5–7] (Figure 1). Stormwater wetlands are designed to provide storage for controlling runoff peaks, flooding, and water quality by various treatments (e.g., settling, bacterial degradation) [8,9]. In the context of this study, the six constructed wetlands also play an important role in ameliorating water

quality before being discharged to the Pike River, impacting ecological health. This warrants an investigation into their function and contribution to the mitigation of stormwater pollution (Figure 1) [5–7]. This investigation is especially needed because these wetlands are between 13 to 20 years old [5–7] and might not have attained the ecological functional maturity in ameliorating water quality as natural wetlands [10]. As such, a careful process needs to be developed that looks into the status of the input pollutants (such as metals and nutrients) from the surrounding watershed into the freshwater constructed wetlands, as well as the possible response from the series of bioindicators in relation to the pollutants, to understand the effect on the biota.

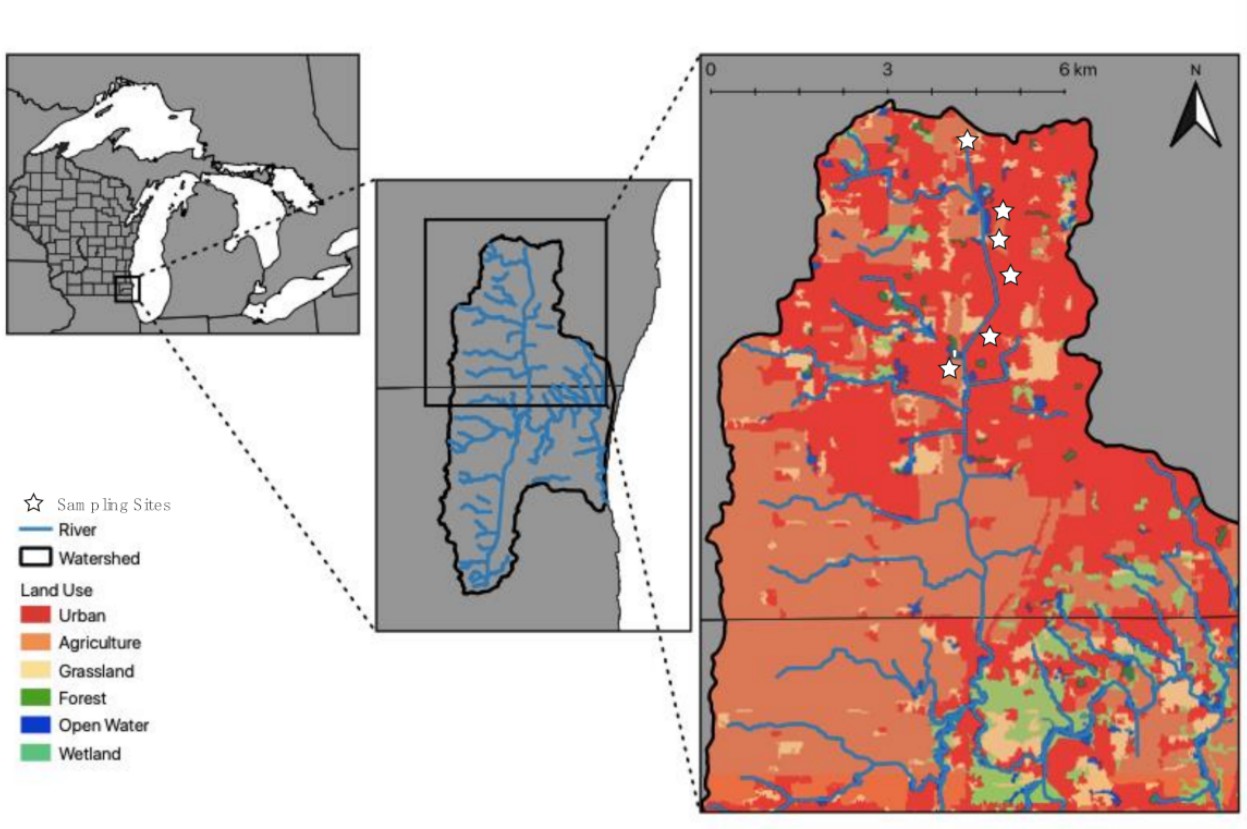

**Figure 1.** Map of the Pike River North Branch (42°43′ N and 87°52′ W) displaying surrounding land use (adapted from Southeastern Wisconsin Regional Planning Commission, SEWRPC 2010). Wetland sampling sites are indicated by white stars and were numbered sequentially from 1 to 6 from north to south.

Nutrients and metals are transported to wetlands through surface flow, precipitation, groundwater, and tides [11], and microbes play an essential role in recycling and removing these elements [11,12]. Wetlands are also highly productive due to active aerobes and anaerobes [13] which quickly recycle nutrients such as nitrate and phosphate [11,12]. Studies have shown that substantial changes can occur in microbial diversity with increasing metal concentrations [14,15]. Metals like zinc (Zn), arsenic (As), mercury (Hg), and lead (Pb) may negatively impact the bacterial community, affecting both diversity and abundance [16–20].

Recent studies illustrated the value of incorporating sediment bacterial assemblage data in monitoring the status of freshwater ecosystems [21,22]. Molecular tools like the 16S rRNA gene sequencing [23] can characterize bacterial communities [24,25] relatively quickly and in detail [24,25]. Several studies have identified sediment communities in freshwater ecosystems using these molecular tools [2,26–29]. Even if the contamination is low, microbial indicators react to change in environmental quality and [30] signal change that might not be detectable in higher trophic level bioindicators such as animals [30]. As

bacterial communities are quickly altered by pollutants such as nutrients and metals, the choice of these bioindicators can give a clear idea about the possible response from the biota of a constructed freshwater wetland ecosystem.

On the other hand, PhytoToxKits[TM] (Microbiotest Inc. 2019, Gent, Belgium) [31], measuring the growth inhibition of indicator plants, has been effective in detecting toxic hazards in reservoir sediments and urban canals subjected to varying levels of nutrients and metal contaminations [32–34]. These constructed wetlands are in a rapidly urbanizing watershed (Figure 1), so bioindicators that have already effectively detected these toxic hazards can be expected to produce a response pattern to compare with [32–35].

These six constructed wetland study sites are on a rapidly urbanized watershed. (Figure 1, Table 1). Other studies indicated the presence of metals [36] and nutrients [37] in urbanizing watersheds. Hence, both these pollutants need to be investigated to see how their presence might affect the associated biota.

**Table 1.** Wetland site, water quality characteristics, and organic matter (OM) percent monitored on ten separate days between June and August 2017 of wetland sites 1–6 in the Pike river watershed (adapted from [35]).

| | | A. Landcover (Percent in Watershed) | | | | |
|---|---|---|---|---|---|---|
| **Wetland Site** | **Watershed Area (ha)** | **Percent Residential** | **Percent Commercial** | **Percent Industrial** | **Percent Agricultural** | **Percent Undeveloped** |
| 1 | 104.45 | 11 | 15.1 | 12.1 | 61.6 | 0 |
| 2 | 334.18 | 42.3 | 0 | 0 | 57.5 | 0 |
| 3 | 267.46 | 41.8 | 0 | 0 | 58.2 | 0 |
| 4 | 2.88 | 58.9 | 6 | 0 | 35.2 | 0 |
| 5 | 493.72 | 15.7 | 14.2 | 20.8 | 0 | 49.3 |
| 6 | 720 | 0 | 72.2 | 20.2 | 0 | 7.2 |

| | | B. Water and Sediment Characteristics | | | | |
|---|---|---|---|---|---|---|
| **Wetland Site** | **Median Temperature (°C)** | **Median pH** | **Median Specific Conductance (mS/cm)** | **Median Dissolved Oxygen (%)** | **Median Dissolved Oxygen (mg/L)** | **Median Organic Matter Percent** |
| 1 | 21.5 | 7.6 | 870 | 105.9 | 9.2 | 8.1 |
| 2 | 21.2 | 7.7 | 634 | 101.8 | 8.7 | 13.3 |
| 3 | 20.2 | 7.2 | 701 | 79.2 | 6.9 | 17 |
| 4 | 19.9 | 7.2 | 907 | 48.1 | 3.8 | 8.4 |
| 5 | 21.4 | 7.7 | 974.5 | 88.6 | 7.8 | 3.8 |
| 6 | 21.8 | 7.1 | 1395.5 | 81.5 | 7.1 | 14 |

This study establishes an approach of using a set of bioindicators (ecotoxicological and bacterial) that are quick, responsive, and have not yet been integrated to understand the effects of pollutants on ecosystem health. These bioindicators can predict a wide range of impacts from pollutants and were applied to understand a wider extent of the ecological health risk from anthropogenic activities in freshwater constructed wetland ecosystems in an urbanizing watershed. First, the study used an ecotoxicological approach with three plant bioindicator species (*Sorghum saccharatum*, *Lepidium sativum* and *Sinapis alba*). It then focused on identifying wetland sediment bacterial taxonomic diversity by 16S rRNA sequencing. Finally, the responses of the bioindicators were correlated to the measured pollutants (nutrients in the water, metals in sediments) present in the constructed freshwater wetlands. Therefore, this study meets the gap of correlating ecotoxicological and bacterial bioindicators with nutrient and metal pollutants as risk assessment measurements in freshwater ecosystems.

The study addresses the research question: does variation in the growth inhibition of plant (*Sorghum*, *Lepidium* and *Sinapis*) bioindicators and sediment bacterial taxonomical diversity correlate and predict response patterns with measured concentrations of nutrient

and metal pollution (i.e., ex post impact indicators for ecological risk assessment) entering wetlands from the surrounding watershed?

## 2. Materials and Methods

### 2.1. Study System, Land Use, and Site Characteristics

This study was conducted in the Pike River Watershed (Racine County, WI, USA). This watershed has been transitioning from agricultural to (sub) urban dominant land uses over the past few decades. Due to this large-scale conversion, flooding became an emergent problem sparking the need for a large-scale restoration project. To address these flood-control issues, a set of wetlands were constructed between 2001 and 2008. Six of these constructed wetlands were selected as the sites for this study (Figure 1) [5–7]. The catchment area and percent land use data of each wetland site were extracted from the Southeastern Wisconsin Regional Planning Commission (SEWRPC) [38] (Table 1). Additionally, the mean water quality characteristics measured in the six wetland sites during summer 2017 are in Table 1.

### 2.2. Sediment Sampling Collection and Water Quality Monitoring

Sediment samples from each wetland site were collected during summer (August) 2015, fall (late August and mid September) 2016, and summer (August) 2017 using a core sampler (5 × 50 cm) and Ekman dredge grab sampler (15 × 15 × 25 cm) [39]. The organic content was measured using 10–15 g of sediment sampled from each wetland site using the loss of weight upon the ignition method [40]

Water quality characteristics (pH, temperature, dissolved oxygen, specific conductance, and turbidity) were monitored at each site using a multi-parameter YSI 6600 sonde [41], and 1 L grab water samples were collected on twelve separate days between June and August 2017 for nutrient measurements. Three water quality readings and water samples were taken along a transect of each wetland site. Water samples for nutrient measurements were stored on ice and transported to the laboratory for analysis within 24 h of sample collection.

### 2.3. Ecotoxicological Assays and Nutrient and Metal Measurements in Wetland Sites

Ecotoxicological tests were carried out following the standard operational procedures for Phytotoxkit[TM] (Microbiotest Inc. 2015, Gent, Belgium) using three plant species: monocot *Sorghum saccharatum* and dicots *Lepidium sativum* and *Sinapis alba* [27,35]. Growth inhibition was measured after 72 h of growth in the sediments collected from each wetland site with respect to their growth in control sediments (washed sand—as provided in the kit) [31,39].

Water samples were analyzed for nitrate and phosphate concentration with a HACH DR 2800[TM] spectrophotometer. Nitrate concentrations were analyzed using the cadmium reduction method with a detection range of 0.3–30.0 mg/L $NO3^-$ [42]. Phosphate concentrations were analyzed using the ascorbic acid method with a detection range of 0.02 to 2.50 mg/L $PO_4^{3-}$ [43]. Nutrient analyses were performed in triplicate for each sample. The mean of three readings was calculated for later analysis.

Sediment samples were tested for the presence and estimated concentration of metals (Ag, Hg, Pb, As, Ni, Zn, and Cd) using X-ray fluorescence (XRF) [44–49]. After large rocks and organic debris were removed, sediment samples from each wetland site were dried at 60–80 °C until a constant weight was obtained. Dried samples were homogenized using a mechanical homogenizer and turned into ~5 g pellets (25 mm diameter and 5 mm height) with a 25-metric ton press pellet. XRF analyses were conducted with a Bruker Tracer III-V+ p-spectrophotometer [44–47] using the red filter to allow x-rays from 14 to 40 KeV to reach the sample. This filter is better for analyzing higher Z elements, such as heavy metals [44–49]. Three readings were taken from each sample. The mean of the three readings was calculated for later analysis.



Calibrations were performed using the National Institute of Standards and Technology [50] standard reference materials (SRMs). These SRMs contained certified amounts of the targeted metals in soil or sediments. The SRMs were obtained from the National Institute of Standard and Technology (NIST). The XRF signal intensity was plotted against the value of each SRM to construct the calibration curves. Blank sample pellets composed of chemically pure silica were used to check for cross-contamination or other interferences. All the analyses were performed in a sample cup under the Si-Pin detector of the Bruker Tracer III-V+ p-spectrophotometer [47–49]. Three readings (in ppm) were recorded for each sample [51]. Finally, the mean of the three readings was calculated for later analysis.

### 2.4. Bacterial Community Structure

DNA was extracted using 0.8 g of each sediment sample with the Fast DNA$^{TM}$ spin kit for soil [52,53]. Three extractions were performed from each sample yielding 70 µL of DNA suspended in DES solution, and 2 µL of the extracted DNA was quantified using a Nanodrop 1000 Spectrophotometer for every extraction [54]. After 72 h, 10–20µL of the extracted DNA (the extraction with the best DNA quantity for each sample) was sent to the University of Wisconsin Madison Biotechnology Centre for library preparation and sequencing of the v3–v4 region in the 16S bacterial rRNA gene using Illumina Next-Generation Sequencing [55].

After the sequences were retrieved electronically, bioinformatics analyses were done using the software Mothur (v1.36.1). This set of analyses used the SILVA database (Release Version 128) for sequence alignment and Greengenes Reference Taxonomy (Version13_8_99) for taxonomy [56]. In Mothur, the sequences were screened to remove any with ambiguous bases longer than 464 bp. Unique sequences were then identified, and duplicates removed. Sequences were aligned as per the SILVA database (Release Version 128), with the start and end of the alignment being specified. After this, the sequences were counted, filtered, and pre-clustered, splitting the sequences by group, sorting them by abundance, listing from most to least abundant, and identifying sequences within 2 nt of each other. Chimera.vsearch was then performed to remove chimeric sequences, and the resulting sequences were classified using Greengenes Reference Taxonomy (Version13_8_99) [56].

### 2.5. Data Analyses

Data distributions were examined for normality and transformed as necessary to meet the assumptions of statistical tests. Count and length data were transformed using a log transformation, while proportional data were transformed using an arcsine transformation [57] prior to statistical analyses [58]. Bacterial taxonomical diversity from each sediment sample collected was calculated using Shannon and Simpson diversity indices for both order and genera [28].

Multifactor analysis of variance (ANOVA) was used to examine the effects of nutrients and metals on plant growth inhibition and bacterial diversity indices, respectively. The effects of measured nutrient concentrations on the plant growth inhibition and bacterial indicators were tested for samples from 2017 when in situ nutrients were measured.

Factor analysis was conducted using the log-transformed concentrations of the metals measured from the wetland sediments [57,58] with maximum likelihood and varimax rotation method based on a correlation matrix. ANOVA was used to measure the metal pollution in ppm on plant growth inhibition and bacterial diversity indices of order and genera. The year was also an independent factor in the model.

Prediction profiles were used together with multi-factor models [58] to examine how values of independent factors (either nutrients or metals) interacted to influence the growth inhibition of plants or bacterial indicators.

Finally, a forward stepwise multiple regression was used to determine the best fit model for the combined predictive linear relationships between pollutants (nutrients and metals) and growth inhibition of the ecotoxicological and bacterial bioindicators. All analyses were performed using JMP$^{®}$ 14 [58].

## 3. Results

### 3.1. Distribution of Measured Nutrients, Metals and Factor Analysis of the Metal Concentrations

The measured nitrate concentrations were between 0 to 11 mg/L, and the phosphate concentrations were between 0 to 1.8 mg/L across all six wetland sites water during summer 2017.

During summer 2015, early fall 2016, and summer 2017, sediment concentration of silver (Ag) ranged from 8 to 13 ppm, arsenic (As) was from 0 to 4 ppm, cadmium (Cd) was from 1.4 to 2.6 ppm, mercury (Hg) was from 0.25 to 2.75 ppm, nickel (Ni) was from 12 to 21 ppm, lead (Pb) was from 0.001 to 0.0035 ppm, and zinc (Zn) was from 5 to 40 ppm across all six wetland sites. Factor analysis was conducted with the metal concentrations resulting in two linear components: Component 1 showed positive loadings for Ag, Zn, As, Cd, and Ni concentrations. Component 2 showed a positive loading for Pb and negative loading for Hg (Figure 2), suggesting that these metals were negatively associated with each other in the sediment samples.

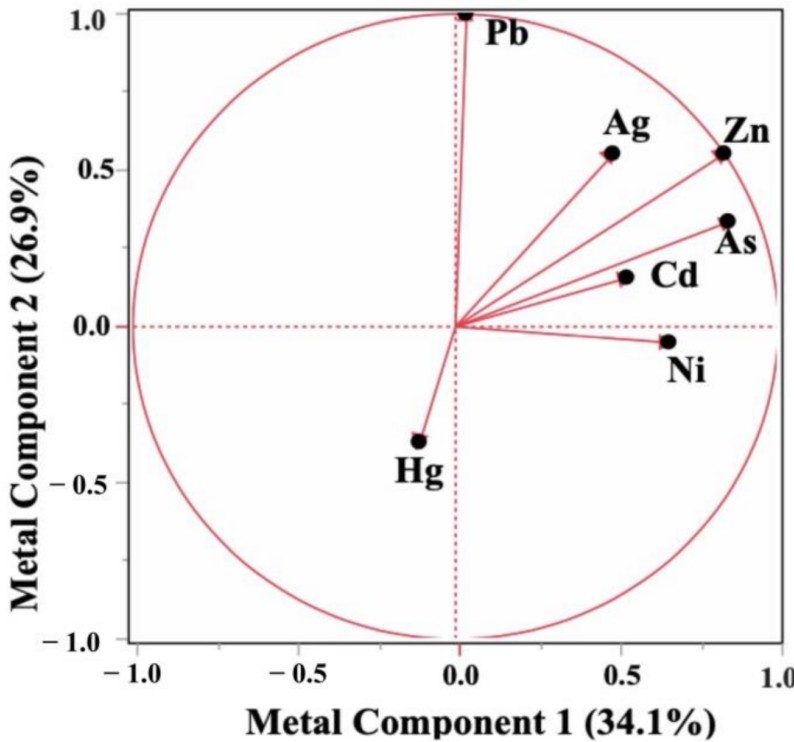

**Figure 2.** Component loadings from multivariate factor analysis using JMP® 14 (SAS 2019). Analysis was conducted on the correlations matrix using the maximum likelihood and varimax rotation method. The XRF detected metals (Ag, Zn, As, Cd, Ni, Pb, and Hg in ppm) were factored into two linear components (Component 1 and 2) which represented 61% of the total variation. Data were from sediments collected from wetland sites 1–6 sampled during 2015–2017. The plot shows rotated factor loadings relative to each component in the multivariate space.

### 3.2. Ecotoxicological Indicators and Their Response to Nutrient and Metal Pollution Stress

The proportions for root and stem growth inhibition values were calculated with respect to growth in control sediments (clean silica sand), with positive values indicating inhibition (reduced growth relative to controls) and negative values indicating stimulation (increased growth relative to controls). For *Lepidium*, root inhibition was in the range of −1.5 to +1.5, and for stem inhibition, − 0.75 to +1.25. For *Sinapis*, root inhibition was in the range of −1.5 to +1.25, and for stem inhibition, −1 to +1.25. For *Sorghum*, the proportion

root inhibition ranged from −1.5 to +1.25, and stem inhibition ranged from −3.5 to +1.5. Responses varied among wetland sites, between years of sampling and by species [39].

Multifactor analysis of variance (ANOVA) was used to examine the effects of measured nutrients on plant growth inhibition. There were different response patterns with respect to nutrients measured directly in the wetlands (Figure 3), where root inhibition was significantly affected by phosphate ($p = 0.0207$) and by its interaction with nitrate ($p = 0.0190$) (Figure 3). Root inhibition increased with higher phosphate concentrations (Figure 3).

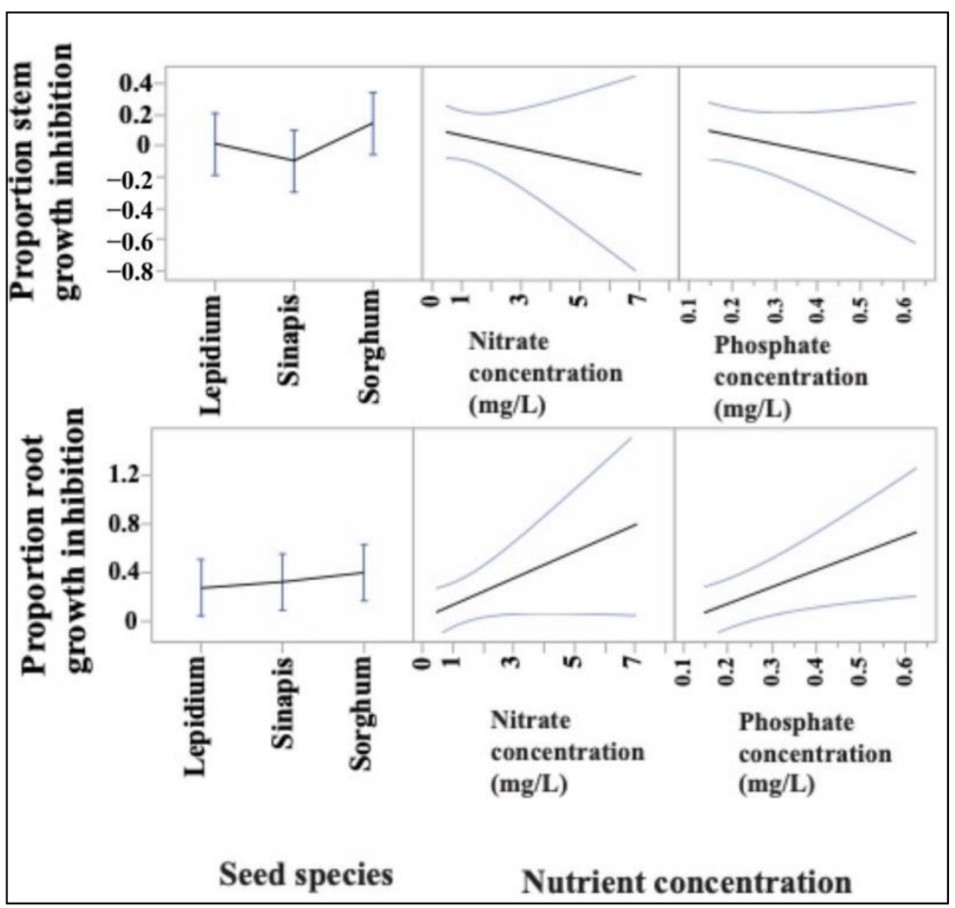

**Figure 3.** Prediction profiles from ANOVA showing the effects of seed species, nitrate and phosphate concentrations (mg/L) measured in wetlands on the growth inhibitions of the bioindicator species *Lepidium sativum*, *Sinapis alba* and *Sorghum saccharatum*. The blue-lined area in each profile represents the 95% confidence prediction interval of the y (continuous) variable). The profiler was set to nitrate at 2.27 mg/L and phosphate at 0.29 mg/L in case of root and stem growth inhibition.

The effects of metal concentrations on ecotoxicological bioindicator growth inhibitions were also examined through ANOVA, using component scores calculated from the factor analysis (Figure 2). None of the metal components had a statistically significant effect on stem growth inhibition with $p$ values for metal component 1 = 0.3511, component 2 = 0.8892, and interaction of components = 0.5697 (Figure 4). However, seed species did respond significantly ($p = 0. 0002$).

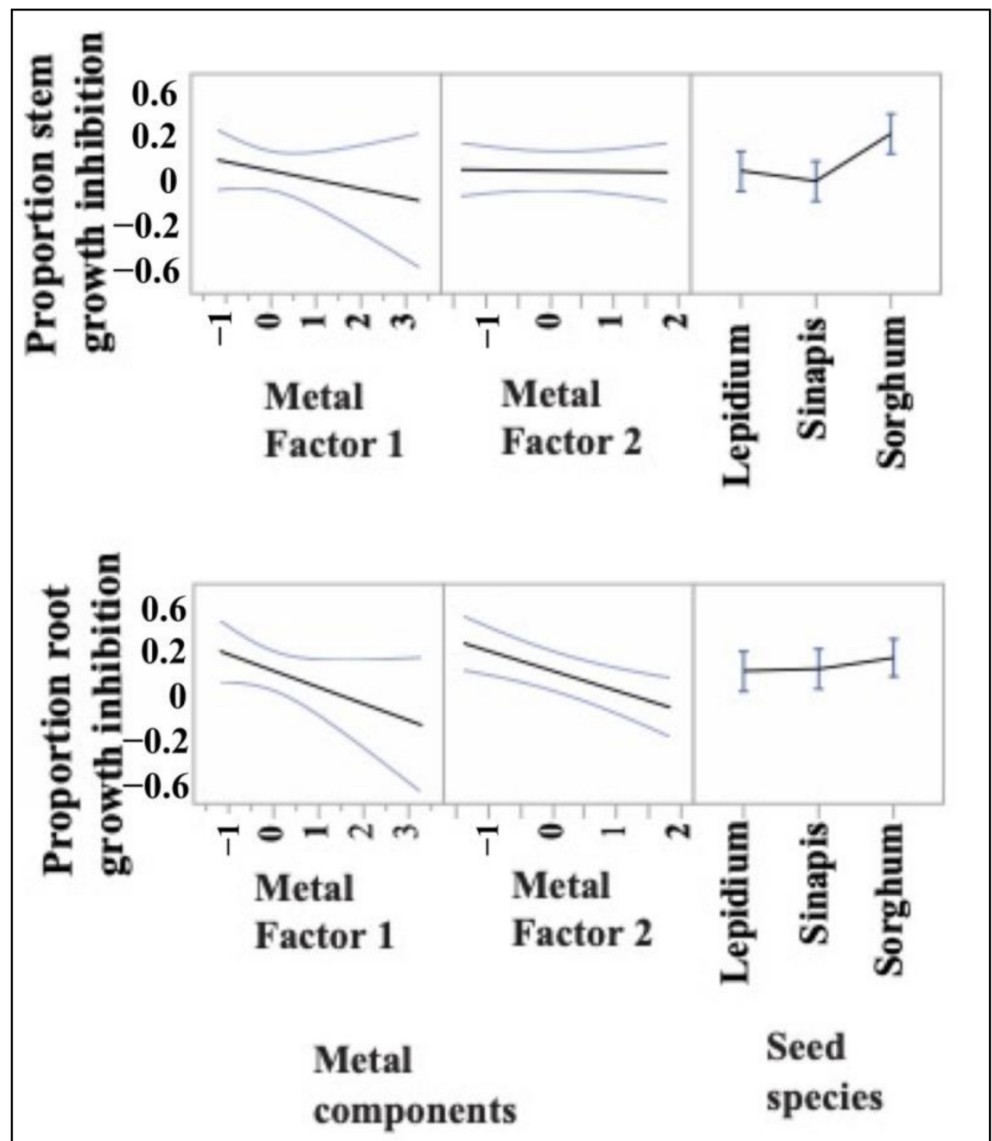

**Figure 4.** Prediction profiles from ANOVA showing the effects of metal components 1 and 2 (from factor analysis) on the growth inhibitions of the bioindicator species *Lepidium sativum*, *Sinapis alba* and *Sorghum saccharatum*. The blue-lined area in each profile represents the 95% confidence interval for the prediction of the response variable. The profiler was set to metal component 1 at 1.0 and metal component 2 at 0.75 in case of root and stem growth inhibition.

While metal pollutants did not have a significant relationship with stem growth inhibition, root inhibition was affected significantly. The positive axis of metal component 2 in the factor analysis was associated with higher Pb and lower Hg concentrations (Figure 2). The negative axis of metal component 2 of the factor analysis was associated with lower Pb and higher Hg concentrations (Figure 2). An increase in metal component 2 (positive axis) showed a significant decrease in root inhibition ($p = 0.0026$, Figure 4). The negative axis of metal component 2 (lower Pb and higher Hg concentrations) was associated with increased root inhibition ($p = 0.0026$, Figure 4).

Finally, a stepwise multiple regression model was calculated to predict the responses of ecotoxicological bioindicators in relation to nutrients (nitrate and phosphate) and metals (Hg and Pb). Each plant bioindicator species was run separately for nutrients (2017 data only) and metals (2015–2017 data). Best fit models using the forward stepping algorithm are presented in Table 2. *Lepidium* stem inhibition exhibited no significant response to variation

in nutrients or metals (Table 2). *Sinapis* stem inhibition had a positive relationship with nitrate but a negative relationship with phosphate, with a total $R^2$ of 0.84 (Table 2). *Sorghum* stem inhibition exhibited a negative association with nitrate but a positive relationship with phosphate. *Sorghum* also had a positive relationship with Pb and Hg with an overall $R^2$ of 0.96 (Table 2). For root inhibition, *Lepidium* growth inhibition was associated positively with nitrate and negatively with Pb and Hg, with an overall $R^2$ of 0.82 (Table 2). *Sinapis* root inhibition was negatively associated with Pb and Hg, with an overall $R^2$ of 0.55 (Table 2). For *Sorghum*, root inhibition had a negative estimated relationship with Pb, with an overall $R^2$ of 0.39 (Table 2).

**Table 2.** Stepwise regression with the estimates of the relationship between nutrient (nitrate and phosphate concentration in mg/L) and metal (Hg, Pb) concentration in ppm with stem and root growth inhibition of *Lepidium sativum*, *Sinapis alba* and *Sorghum sachharatum* in wetland sites 1–6, along with appropriate $R^2$ and *p* values. The blank spaces revealed no estimates of relationship.

| | Seed Species | Nitrate | Phosphate | Pb | Hg | $R^2$ |
|---|---|---|---|---|---|---|
| (A) Stem Inhibition | | | | | | |
| | *Lepidium* | | | | | |
| | *Sinapis* | 0.31 (*p* = 0.0656) | −1.96 (*p* = 0.0286) | | | 0.84 |
| | *Sorghum* | −1.77 (*p* = 0.1249) | 1.88 (*p* = 0.2417) | 998.83 (*p* = 0.1879) | 4.80 (*p* = 0.1548) | 0.96 |
| (B) Root Inhibition | | | | | | |
| | Seed Species | Nitrate | Phosphate | Pb | Hg | $R^2$ |
| | *Lepidium* | 2.12 (*p* = 0.0998) | | −1559.8 (*p* = 0.1216) | −6.75 (*p* = 0.1310) | 0.82 |
| | *Sinapis* | | | −1011 (*p* = 0.1962) | −2.46 (*p* = 0.1196) | 0.55 |
| | *Sorghum* | | | −234.54 (*p* = 0.0031) | | 0.39 |

### 3.3. Sediment Bacterial Bioindicators and Response to Nutrient and Metal Pollution

A total of 67,503 sequences were identified from all wetland sites. The lowest number of sequences identified in a sample was 115, while the highest was 16,005. At the broadest level, 261 unique orders and 924 unique genera were identified. These unique orders and genera were used to calculate the bacterial indicators (Shannon and Simpson diversity indices of order and genera) across all six wetland sample sites. Across all sites and sampling times, the Shannon diversity index of orders ranged between 1.38 and 4.46 and the Simpson diversity index of orders ranged from 0.46 to 1.90. The Shannon diversity index of genera ranged between 1.30 and 5.18 and the Simpson diversity index of genera ranged from 0.64 to 0.99.

Some of the most abundant orders across all wetland sites and sampling times (summer 2015, fall 2016, and summer 2017) included: Bacillales, Bacteroidales, Clostridiales, Actinomycetales, Burkholderiales, Rhizobiales, Pseudomonadales, and Xanthomonadales (Table A1, Appendix A). Some of the most abundant genera across all wetland sites and sampling times (summer 2015, fall 2016, and summer 2017) included: *Bacillus, Clostridium, Pseudomonas, Flavobacterium, Treponema, Thiobacillus, Crenothrix, Streptococcus, Bdellovibrio,* and *Pelomonas* (Table A1, Appendix A).

The presence of potential bias in the diversity indices related to the number of 16S rRNA gene sequences retrieved was analyzed for each of the wetland sites. The slope values for the following relationships with respect to the total number of sequences retrieved were calculated: Shannon diversity indices of order (slope = 0.9132, $R^2$ = 0.3661, *p* = 0.0218), Simpson diversity indices of order (slope = 0.0848, $R^2$ = 0.1492, *p* = 0.6159), Shannon diversity indices of genera (slope = 0.443, $R^2$ = 0.0701, *p* = 0.3602), and Simpson diversity indices of genera (slope = 0.0557, $R^2$ = 1597, *p* = 0.1568) (Figure A1, Appendix B).

Multifactor analysis of variance (ANOVA) was used to examine the effects of measured nutrients on bacterial diversity indices. The effects of measured nutrient concentrations in the wetlands on bacterial diversity indices are shown in Figures 5 and 6. The Shannon diversity of order decreased significantly with increasing phosphate ($p < 0.0001$) and nitrate ($p = 0.0299$) concentrations (Figure 5). The nutrient interaction effect ($p = 0.0032$) was also significant on the Shannon diversity of order. The Simpson diversity index of order decreased significantly with increasing nitrate ($p < 0.0001$) and phosphate concentrations ($p < 0.0001$). A strong interaction effect of nutrients ($p < 0.0001$) (Figure 5) was observed on the Simpson diversity index of order.

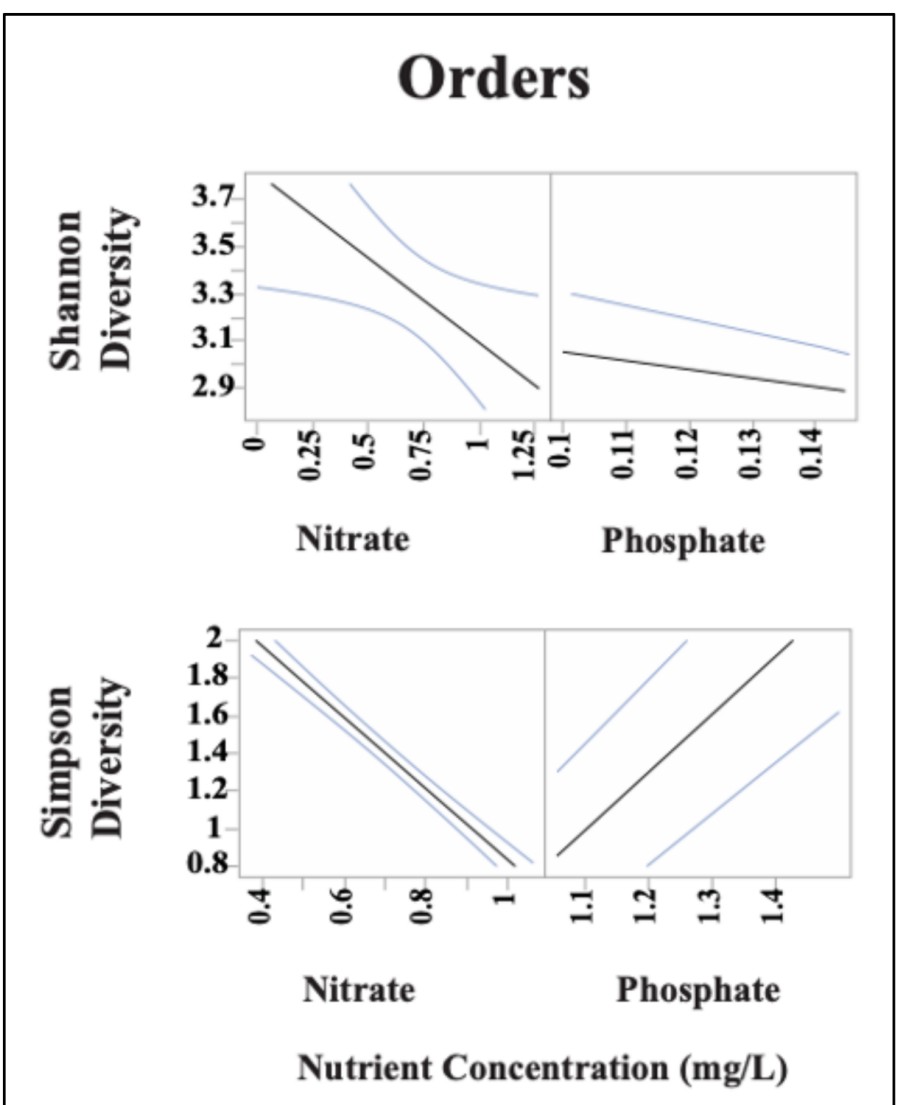

**Figure 5.** Prediction profiles from ANOVA, showing the effect of measured nitrate and phosphate concentration (mg/L) on the Shannon and Simpson diversity indices identified as taxonomic order out of wetland sites 1–6 during summer 2017. The blue-lined area in each profile represents the 95% confidence prediction interval of the response variable. The profiler was set for median ambient conditions, with 2.19 nitrate at mg/L and phosphate at 0.27 mg/L in the case of Shannon and Simpson diversity, respectively.

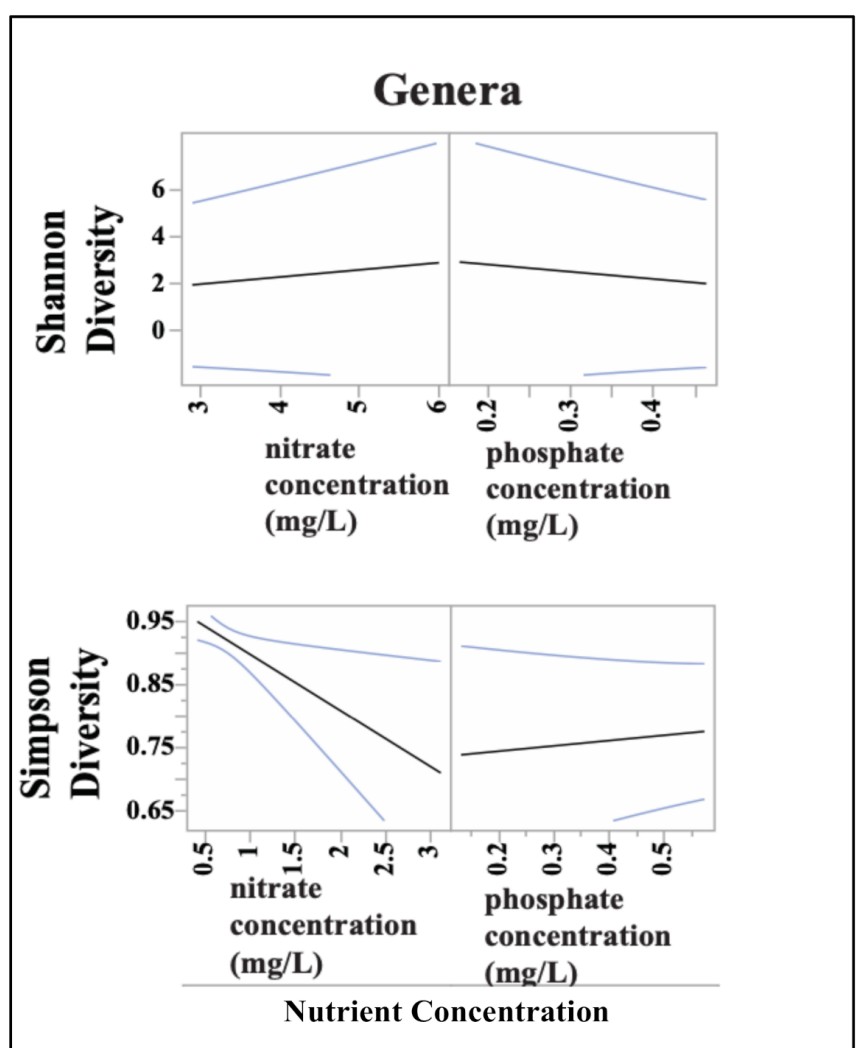

**Figure 6.** Prediction profiles from ANOVA, showing the effect of measured nitrate and phosphate concentration (mg/L) on the Shannon and Simpson diversity indices identified as taxonomic genera out of wetland sites 1–6 during summer 2017. The blue-lined area in each profile represents the 95% confidence prediction interval of the response variable. The profiler was set for nitrate at 2.72 mg/L and phosphate at 0.49 mg/L in the case of Shannon diversity, and nitrate at 2.60 mg/L and phosphate at 0.32 mg/L in the case of Simpson diversity.

For bacterial genera, the Shannon diversity index of genera decreased significantly with increasing phosphate concentrations ($p = 0.0266$). However, no significant effect was observed for nitrate ($p = 0.7033$) or nutrient interactions ($p = 0.3604$) (Figure 6). The Simpson diversity index of genera was observed to decrease significantly with increasing nitrate ($p = 0.0215$). No significant effect of phosphate ($p = 0.1673$) was observed on the Simpson diversity index of genera. A strong interaction effect of nutrients ($p = 0.0087$) (Figure 6) was seen on the Simpson diversity index of genera.

The effects of metal concentrations on bacterial diversity indices parameters were examined through ANOVA, using component scores calculated from the factor analysis (Figure 2). The metal component 1 (increased concentrations of As, Ag, Cd, Ni, and Zn in ppm) had a significant effect on the Shannon diversity index of order ($p = 0.0801$), and the Simpson diversity index of order ($p < 0.0001$) (Figure 7). With an increase in metal component 1, an increase in the Shannon diversity index of order and decrease in the Simpson diversity index of order was observed (Figure 7). By contrast, an increase in metal component 2 (interpreted as associated higher Pb and lower Hg concentrations) was

associated with an increase in the Shannon diversity index of order ($p < 0.0001$), and the Simpson diversity index of order ($p = 0.0085$) (Figure 7). The interaction effect of metal components 1 and 2 had a significant effect on both the Shannon ($p < 0.0001$) and Simpson ($p = 0.0084$) diversity indices of order (Figure 7).

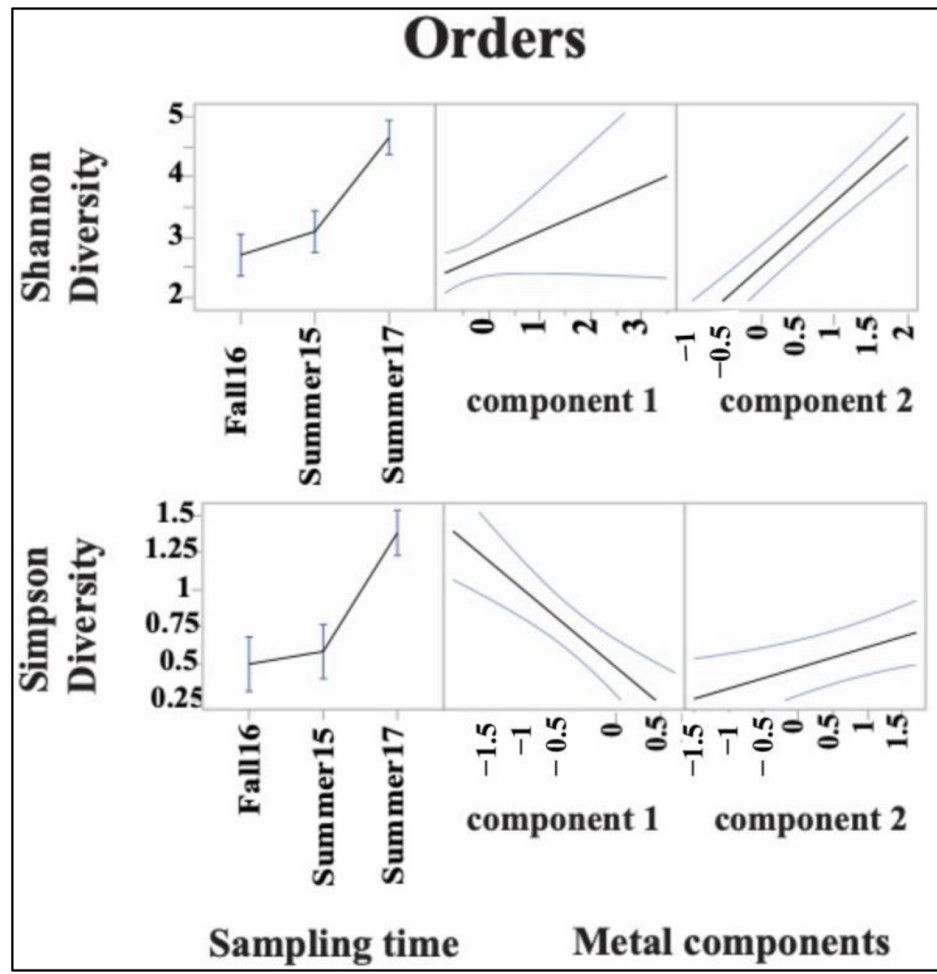

**Figure 7.** Prediction profiles from ANOVA showing the effect of metal component 1 and 2 (from principal component analysis) on the Shannon and Simpson diversity indices identified as taxonomic order from wetland sites 1–6 during fall 2016 and summer 2015 and 2017. The blue-lined area in each profile represents the 95% prediction confidence interval of the response variable. The profiler was set to metal component 1 at −0.0516 and metal component 2 at 0.1825 in the case of Shannon and Simpson diversity, respectively.

Metal component 1 did have a significant effect on the Shannon diversity index of genera ($p = 0.0413$), but not on the Simpson diversity index of genera ($p = 0.0945$) (Figure 8). With an increase in metal component 1, the Shannon diversity index of genera decreased (Figure 8). The Simpson diversity index of genera ($p < 0.0001$) increased with the increase in metal component 2 (interpreted as associated higher Pb and lower Hg concentrations), but no significantly relationship was seen with the Shannon diversity index of genera ($p = 0.5952$) (Figure 8). The statistical interactions among metal components 1 and 2 also had a significant effect on the Simpson diversity ($p < 0.0001$) but not on the Shannon diversity of genera ($p = 0.072$).

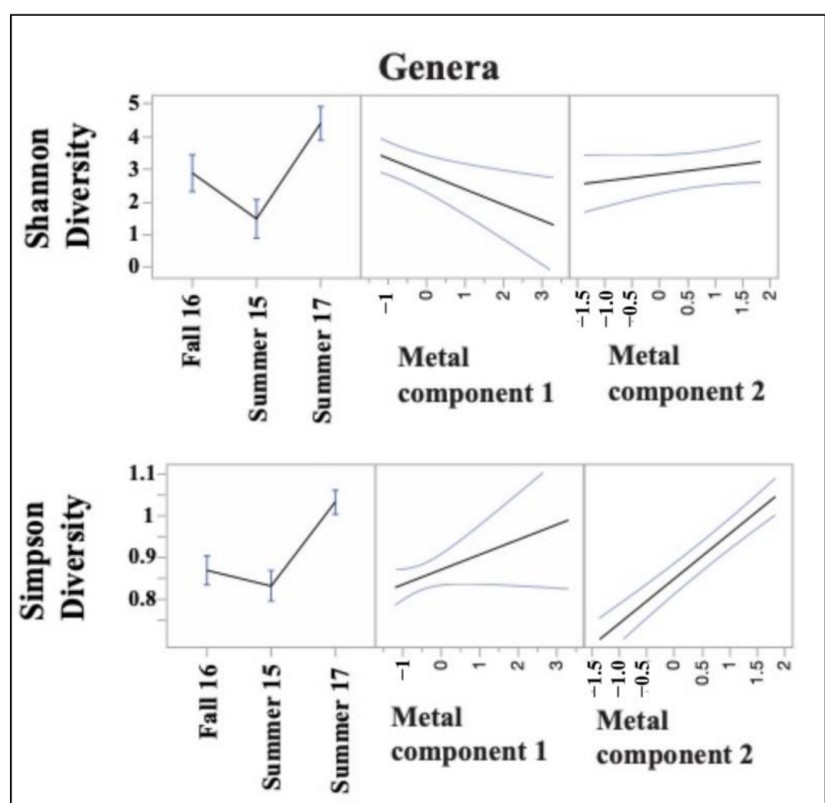

**Figure 8.** Prediction profiles from ANOVA showing the effect of metal component 1 and 2 (from principal component analysis) on the Shannon and Simpson diversity indices identified as taxonomic genera from wetland sites 1–6 during fall 2016 and summer 2015 and 2017. The blue-lined area in each profile represents the 95% prediction confidence interval of the response variable. The profiler was set to metal component 1 at 0.65 and metal component 2 was set to 0.18 in the case of both Shannon diversity and Simpson diversity.

Finally, a stepwise multiple regression model was calculated to predict the responses of bacterial bioindicators in relation to nutrients and metals. The metals Pb, Hg, As, and Zn concentrations were selected for multiple regression analysis due to their high loadings in the metal factor analysis (Figure 2). The Shannon diversity index of order was associated positively with Hg and Pb (Table 3, $R^2 = 0.33$). The Shannon diversity index of bacterial genera was associated positively with Hg and As and negatively with Pb ($R^2 = 0.52$, Table 3). The Simpson diversity index of genera was associated positively with Hg and Pb ($R^2 = 0.27$, Table 3).

**Table 3.** Stepwise regression with the estimates of relationship between metals concentrations with the highest loadings from factor analysis (As, Zn, Hg, Pb) regressed against the Shannon diversity index, Simpson diversity index and total number of identified phyla and genera in wetland sites 1–6, including *p*-values and total $R^2$ for the regression.

| | **As** | **Zn** | **Hg** | **Pb** | $R^2$ |
|---|---|---|---|---|---|
| (A) Order Shannon diversity index | | | 3.76 ($p = 0.0084$) | 1794.42 ($p = 0.00169$) | 0.33 |
| (B) Order Simpson diversity index | | | | | 0 |
| | **As** | **Zn** | **Hg** | **Pb** | $R^2$ |
| (D) Genera Shannon diversity index | 5.53 ($p = 0.00022$) | | 11.0024 ($p = 0.00001$) | −1845.24 ($p = 0.0066$) | 0.52 |
| (E) Genera Simpson diversity index | | | 0.36 ($p = 0.01618$) | 161.35 ($p = 0.00595$) | 0.27 |

## 4. Discussion

The study investigated the correlation and specific response patterns of plants' bioindicator growth inhibition and sediment bacterial taxonomic diversity with nutrients and metals entering into freshwater constructed wetlands from the surrounding watershed.

### 4.1. Specific Response Patterns of Ecotoxicological Bioindicators to Nutrient and Metal Pollution Stress

Factor analysis revealed significant effects of associated groups of metals (component 2: Pb and Hg) (Figure 2). Especially in the negative axis of metal component 2, the combination of decreasing Pb and increasing Hg (Figure 2) was associated with increased root inhibition (Figure 4). The regression analysis also suggested a negative estimated relationship between Hg, a chemical of concern often released into the environment through industrial pollution, mining, and the burning of fossil fuels [53,54], and the root growth inhibition of two of the three plant bioindicator species (Table 2 and Figure 4). Experiments have shown that Hg causes reduced growth in the root and stem of seedlings [59] due to the production of reactive oxygen species (ROS), causing damage in the cell membranes, chloroplast pigments, and nucleic acids [60,61].

In factor analysis, the root growth inhibition of plant bioindicators decreased in association with increased Pb (Figure 4), but in regression analysis, a negative estimated relationship with *Lepidium, Sinapis, Sorghum*, and Pb was observed (Table 2). This suggests that the metal factor analysis alone was not able to depict the individual interplay of the Pb concentration with the plant indicators clearly, and that different statistical tests can reveal a more robust risk assessment.

Agricultural and residential land uses produce runoff rich in nutrients such as phosphate and nitrate due to fertilizers and pesticides in lawns, gardens, and agricultural fields [62]. Our results indicate that high phosphate is associated with higher growth inhibition (Figure 3), which was surprising since phosphate deficiency reportedly causes growth inhibition in plants [63]. This suggests that there were interactions between nutrients (e.g., from fertilizers) and metals (e.g., from pesticides). This became evident in the regression analysis, where phosphate did not have a significant association with the root growth inhibition of the plant bioindicators when analyzed individually (Table 2).

The metal component 1 of the factor analysis had positive loadings for an array of metals Ni, Cd, As, and Zn (Figure 2), which had a modest impact on growth inhibition (Figure 4). Our previous studies performed on textile-dye-contaminated soils in India [35] demonstrated that a combination of metals present in a soil ecosystem could affect the plant bioindicator species. This study's results are consistent with a combination of metals interacting to have a toxic effect on bioindicator plant species [35].

The regression analyses for each plant bioindicator also revealed different relationships between inhibition and facilitation when analyzed for the effects of nutrient and metal pollution (Table 2). The reasons contributing to different responses by different ecotoxicological bioindicator species are grounds for further study. Herbicides and metals are well-known to affect the growth and development of *Sorghum* [64,65]. In comparison to *Sinapis* and *Lepidium,* the *Sorghum* frequently exhibited negative inhibition (facilitation) in this study.

### 4.2. Bacterial Bioindicators

Analyses of extracted DNA from sediment collected from the six wetlands over the study period demonstrated a large variety in the bacterial community assemblages. The most commonly found orders across all sampling sites over time were: Bacillales, Bacteroidales, Clostridiales, Actinomycetales, Burkholderiales, Rhizobiales, Pseudomonadales, and Xanthomonadales (Table A1, Appendix A). These results are consistent with other published studies concerning bacterial community diversity in wetland soils or sediment [66–72].

The Shannon and Simpson diversity indices were used in this study to calculate the diversity matrices of the bacterial population. It is advisable to include multiple indices when attempting to characterize differences in community composition [73]. The use of various indices avoids bias towards the richness or abundance of the identified operational taxonomic units (OTUs). The Shannon and Simpson diversity indices have been used previously for characterizing bacterial communities [5], and each provides somewhat different information. The Shannon diversity index is more sensitive to the taxonomic richness, resulting in identifying each unique OTU, adding to the index value evenness [73]. By comparison, the Simpson diversity index is weighted towards the most abundant OTU of a sample [73,74]. Moreover, in this study, the wide range of variation of the indices for both order and genera across all the wetland sites demonstrated variations in bacterial community assemblages—a primary consideration when selecting good sub-metrics for indicators [75].

It is also essential to recognize if the number of sequences detected influences the diversity indices. In this study, the correlations between the total number of sequences in the sample and Shannon diversity index of order were only significant ($p = 0.0218$) with an $R^2$ value of 0.36 (Figure A1, Appendix B). This suggests that the number of sequences found in the wetland samples affected the order diversity more than the genera. However, with the low $R^2$ value found between the total number of sequences in the sample and the Shannon diversity index of order as stated indicates that a relationship is not strongly present.

Generally, the Shannon diversity index ranges between 1.5 and 3.5, while the Simpson diversity index ranges from 0 to 1 [76]. In this study, the Shannon diversity indices of order and genus were 1.38–4.46 and 1.30–5.18, respectively (Figure A1, Appendix B). Other studies reported the Shannon diversity indices ranging between 3.57 and 5.38 [7,77] (Figure A1, Appendix B). The Simpson diversity index of order and genera ranged between 0.8 and 1 (Figure A1, Appendix B), in this study. Other studies reported the Simpson diversity indices between 0.46–1 [77,78]. The value of Simpson diversity was mainly in the higher range; this shows the applicability of the Simpson diversity index more in this study compared to the Shannon.

Specific Response Patterns of Bacterial Bioindicators to Nutrient and Metal Pollution Stress

This study detected an array of significant relationships between nutrients and sediment bacterial communities. The Simpson diversity of order and genera were observed to decrease with increasing measured nitrate concentration (Figures 5 and 6). Fertilizers and pesticides from surrounding agricultural and residential lands are rich in phosphate and nitrate, serving as a source of nutrients that may run off into these wetland sites [79,80]. There are interactions between the loadings of nutrients (e.g., from fertilizers) and loadings of other pollutants, such as metals associated with pesticides, causing less bacterial diversity.

Studies have shown that microbial diversity can substantially change in response to increasing metal concentrations [15,16], even leading to extinction [17]. Examination of metal component 2 suggests that the negative axis with metal component 2 (decreasing Pb and increasing Hg) (Figure 2) was associated with reduced bacterial diversity (Figures 7 and 8). Studies indicate that most heavy metals, including Hg, can cause toxic effects on bacterial cells at low concentrations [81,82]. However, the multiple regression analysis showed that Pb, Hg, As, and Zn concentrations had a significant positive relationship with the bacterial diversity indices in certain instances (Table 3). Bacterial communities have shown the ability to develop metal resistance when exposed for long durations, resulting in a positive relationship of bacterial indicators with concentrations of metals such as Pb and Hg [83–90]. The results of this study suggest that this may be occurring in the constructed wetlands. The metal component analysis revealed the effect of interactions of metals like Pb and Hg on bacterial indicators. In contrast, the regression analysis showed how the metals (Pb and Hg) affected the bacterial indicators when analyzed individually (but concurrently with metals like As and Zn).

One of the challenges for monitoring environmental impacts is identifying and developing indicators that can capture and integrate the effects of pollutants or stressors across various (sometimes mismatched) spatial and temporal scales. For example, chronic stressors such as baseline nutrient loading from agricultural fields provide different signals than acute events such as a manure spill or pesticide application, whose detection by direct chemical measurement may be missed between monitoring sessions. To this point, robust multi-metric indicators must be constructed to include an array of biological sub-metrics that can detect biological responses to human activities across spatial and temporal scales [75]. The situation is made more complicated because interactions among different stressors in nature may result in complex response patterns that can result in the interpretation of the patterns detected being very context-dependent.

In this regard, prediction profiles provide a valuable tool for visualizing the complexity of interactions among pollutants and understanding why a single relationship for a single indicator is not sufficient in characterizing a biological response signature. An example demonstrating the interaction effect of metal factors on root inhibition is shown in Figure 9. The top half shows the expected relationship between root inhibition and metal component 1 when metal component 2 is set to a value of 1.5, which would indicate high levels of Pb and low Hg (Figure 9A). In this case, the prediction is that one would expect to see at most a small positive effect, if any, of the increasing levels of metal component 1 on root inhibition. By contrast, the prediction profile shown in Figure 9B illustrates the predicted changes in root inhibition relative to changing levels of metal component 1 when the level of metal component 2 is held to -1.0 (low Pb with high Hg). In this circumstance, the slope of the relationship between root inhibition and metal component 1 is negative, where increasing levels of metal component 1 are predicted to result in lower levels of root inhibition (Figure 9B).

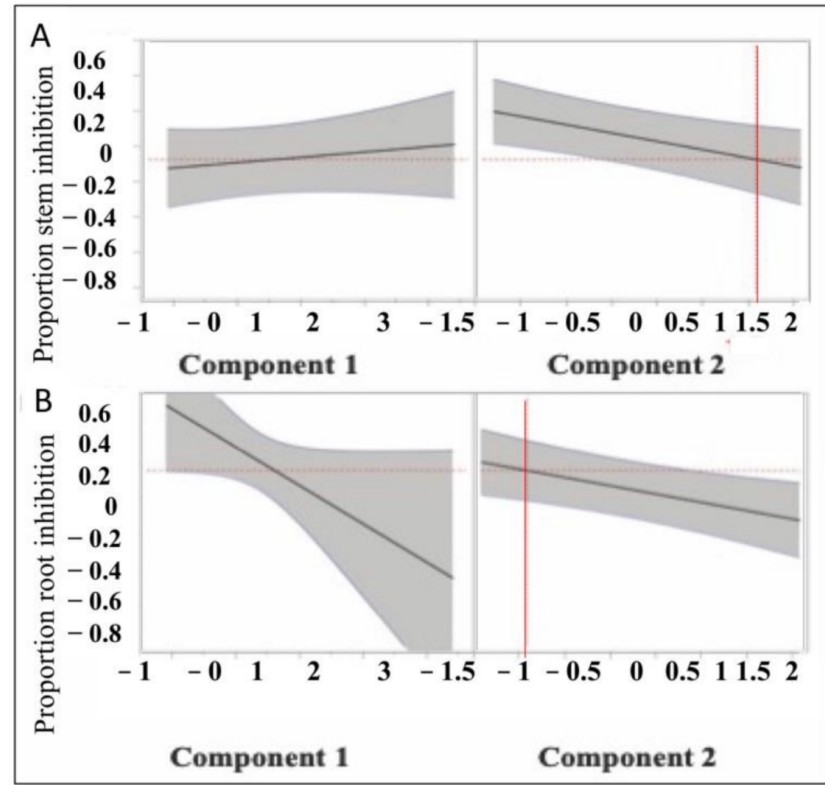

**Figure 9.** Prediction profiles for the effects of metal factor 1 and metal factor 2 on root inhibition. (**A**) Profile when metal component 2 is set to 1.5, and (**B**) profile when metal component 2 is set to −1.0 (high Hg and low Pb).

The results from this study demonstrate that interactions among complex mixtures of nutrients and metals in wetland ecosystems can produce relationships with bioindicators that run counter to the predictions made by considering individual pollutants independently [91]. For example, with the increase in Pb, a decrease in root inhibition was observed in the factor analysis (Figure 4). However, a negative relationship in the regression analysis was observed between the plant bioindicators and Pb concentration (Table 2). For bacterial diversity, the detected Hg, even at low concentrations, showed decreasing Shannon and Simpson diversity indices in the factor analysis (Figures 7 and 8), but a positive relationship was found in the regression analysis between bacterial bioindicators (Shannon and Simpson diversity indices) and Hg concentration (Table 3). The factor analysis showed a combined effect of the pollutants, whereas the regression analysis showed individual effects. All these conflicting trends open up a wide scope for further investigations. The effect of ecotoxicological and bacterial bioindicators needs to be thoroughly investigated in further studies with respect to the response to the pollutants (nutrients and metals) to establish a specific usage method. A controlled study with microcosms of these wetland sites, where the effects of pollutants (nutrients and metals) can be tested repeatedly on the bacterial and ecotoxicological bioindicators, can be very useful [66,92]. It also needs to be reiterated that these constructed wetlands are not very old [5–7]. Hence, these bioindicators can be applied and compared with the results found in natural wetlands. Although these comparisons were not under the scope of the present study, they may shed light on the contradictions found.

This study showed how disparate bioindicators (ecotoxicological and bacterial), which are quick, responsive, and well-established in the literature, can provide a clearer picture of ecological risk in constructed wetlands. The bacterial bioindicators showed us the immediate effect of a pollutant (for example, reduction in bacterial diversity with respect to Hg), whereas plant bioindicators can show an impact after the pollutants have entered higher trophic levels. Therefore, these bioindicators are capable of predicting a wide range of impacts from pollutants and provide correlations and response patterns in relation to pollution-related stressors (such as nutrients and metals) entering the wetlands from the surrounding watershed. There are some contradictions in the results that can be investigated in future studies before these sets of bioindicators are clearly established as tools for measuring ecological integrity in freshwater constructed wetlands.

## 5. Conclusions

The ecotoxicological and bacterial taxonomical diversity bioindicators under investigation in this study demonstrated a clear correlation and specific predictive trends when analyzed with measured watershed pollutants, such as nutrients and metals detected in the wetland sediments and water. Hence, these bioindicators can serve as predictive bioindicators for ecological risk assessment of freshwater wetlands.

**Author Contributions:** T.J.E. assisted in the study conceptualization, design, data analysis, and funding of the project. The majority of the laboratory work was conducted in T.J.E.'s lab. C.F.W. contributed to the conceptualization and mentorship of the project and editing the manuscript. The microbial work was performed in C.F.W.'s laboratory. S.A.M. assisted in the field work, and with writing and editing the manuscript. S.G.R. did the conceptualization and field work of the project with the help of T.J.E., C.F.W. and S.A.M. S.G.R. also planned and conducted all the lab work, did the data collection, curation and analysis, did the original draft preparation, and the writing, reviewing and editing of the manuscript of the project. All authors have read and agreed to the published version of the manuscript.

**Funding:** Funding for this work was provided by The Village of Mount Pleasant (Racine, WI, USA) and the Wm. Collin Kohler's Foundation Sustainable Peacebuilding Fund at UW-Milwaukee.

**Institutional Review Board Statement:** Not applicable.

**Informed Consent Statement:** Not applicable.

**Data Availability Statement:** Data about the landuse patterns of the wetland sites (1–6) can be referred from https://www.sewrpc.org/SEWRPC.htm (accessed on 14 March 2015). The Bioinformatics software used for microbial data analysis can be referred from https://mothur.org/ (accessed on 14 March 2015).

**Acknowledgments:** We want to thank The Village of Mount Pleasant (Racine, WI, USA) and Wm. Collin Kohler's foundation for supporting this study.

**Conflicts of Interest:** The authors declare no conflict of interest.

# Appendix A

**Table A1.** The relative abundance of some of the most abundant bacterial orders and genera identified in the collected wetland sediments during summer 2015, fall 2016, and summer 2017, identified by 16S rRNA gene sequencing in wetland sites 1, 2, 4, 5, and 6.

| Genera | Summer 15 | Fall 16 | Summer 17 | Order | Summer 15 | Fall 16 | Summer 17 |
|---|---|---|---|---|---|---|---|
| *Bacillus* | 60.30 | 3.85 | 0.04 | Rhodospirillales | 3.24 | 1.68 | 4.56 |
| *Clostridium* | 16.86 | 18.62 | 18.11 | Vibrionales | 9.60 | 0.00 | 0.00 |
| *Pseudomonas* | 3.19 | 26.24 | 41.67 | Saprospirales | 6.64 | 3.22 | 12.03 |
| *Streptococcus* | 2.20 | 0.71 | 2.00 | Bacillales | 59.49 | 7.33 | 3.39 |
| *Bdellovibrio* | 4.70 | 0.00 | 25.90 | Bacteroidales | 5.74 | 2.46 | 15.80 |
| *Flavobacterium* | 5.97 | 5.29 | 11.84 | Caldilineales | 3.36 | 1.84 | 3.10 |
| *Treponema* | 3.60 | 2.17 | 12.22 | Clostridiales | 15.24 | 10.91 | 17.97 |
| *Thiobacillus* | 5.54 | 8.52 | 16.04 | Myxococcales | 8.37 | 2.97 | 9.84 |
| *Paenibacillus* | 2.81 | 0.00 | 0.00 | Pirellulales | 5.01 | 2.60 | 6.20 |
| *Gemmata* | 2.86 | 0.00 | 0.00 | Actinomycetales | 20.39 | 5.90 | 12.09 |
| *Vibrio* | 13.04 | 0.00 | 0.00 | Burkholderiales | 14.25 | 101.86 | 52.31 |
| *Pelomonas* | 0.00 | 47.62 | 5.31 | Rhizobiales | 9.86 | 6.92 | 17.27 |
| *Herbaspirillum* | 0.00 | 20.66 | 2.39 | Xanthomonadales | 6.81 | 7.19 | 10.53 |
| *Geobacter* | 1.89 | 5.61 | 3.50 | Desulfuromonadales | 0.79 | 8.74 | 1.41 |
| *Gaiella* | 1.41 | 4.35 | 0.00 | Gaiellales | 2.64 | 11.71 | 2.84 |
| *Sphingomonas* | 0.00 | 11.33 | 10.13 | Rhodocyclales | 2.67 | 4.38 | 9.12 |
| *Ralstonia* | 0.00 | 3.49 | 0.34 | Pseudomonadales | 3.69 | 9.62 | 18.59 |
| *SJA-88* | 0.00 | 2.01 | 11.42 | Rhizobiales | 9.86 | 6.92 | 17.27 |
| *Rhodobacter* | 0.00 | 0.00 | 5.78 | Sphingomonadales | 2.43 | 6.29 | 15.34 |
| *Hyphomicrobium* | 1.76 | 0.00 | 4.81 | Bdellovibrionales | 1.69 | 0.68 | 10.97 |
| *Crenothrix* | 2.90 | 2.17 | 10.26 | Fusobacteriales | 0.62 | 0.00 | 6.12 |
| *Methylotenera* | 1.50 | 4.74 | 10.29 | Rhodocyclales | 2.67 | 4.38 | 9.12 |
| | | | | Flavobacteriales | 3.79 | 2.81 | 6.92 |
| | | | | Rhodobacterales | 3.36 | 1.58 | 6.48 |
| | | | | Pirellulales | 5.01 | 2.60 | 6.20 |
| | | | | Myxococcales | 8.37 | 2.97 | 9.84 |

**Appendix B**

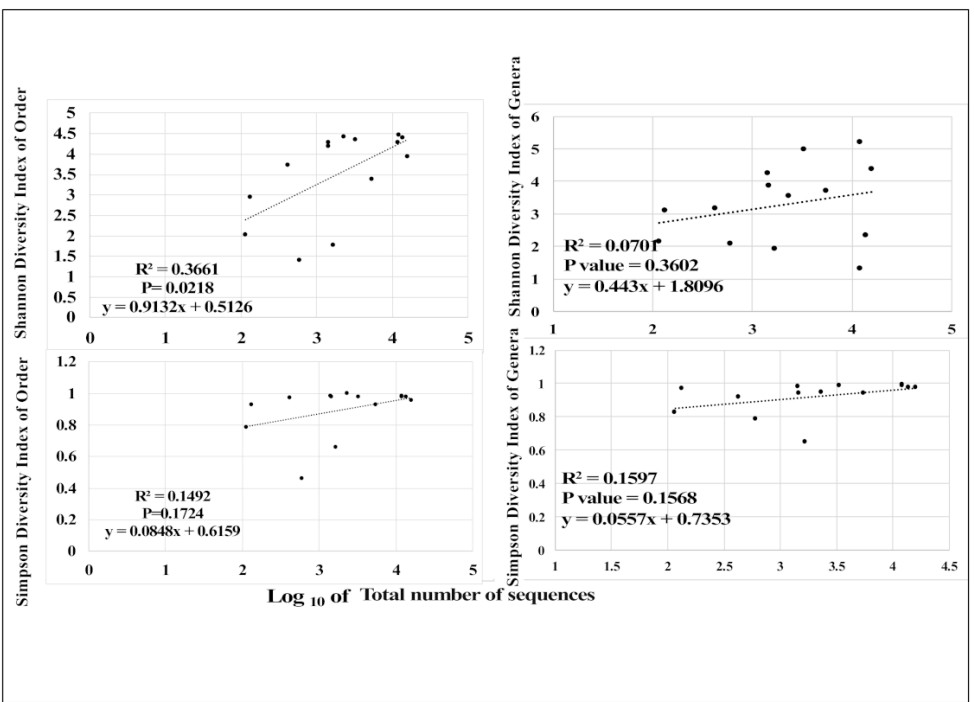

**Figure A1.** Scatterplot and line of fit for the Shannon and Simpson diversity indices of order and genera identified by 16S rRNA gene sequencing in samples from wetland sites 1–6. Sequence numbers are shown as $Log_{10}$ values.

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
