# Peer review of "Effects of Land Use and Pollution Loadings on Ecotoxicological Assays and Bacterial Taxonomical Diversity in Constructed Wetlands"

_diversity, doi:10.3390/d13040149_

Round 1

Reviewer 1 Report

This research is an interesting approach to determine the potential ecosystem harm associated with complex mixtures of contaminants in freshwater wetlands.  Although no new specific methods were used to examine the combined effects of metals and nutrients, the combination of two disparate indicators (plant growth inhibition and the microbial community) appears to be novel and potentially valuable in ways that could lead to a better understanding of contaminant risk.  How best to examine complex mixtures is a difficult problem that requires new thinking; the approach used here may prove insightful.  However, like any new indicator, additional research will be needed before the meaning and full understanding of effects measured by these indices will be known.  This report raises many questions about the efficacy of the indicators used, especially given some contradictory results, and should stimulate additional research to refine the technique.

Some major comments:

The authors do not address differences between restored and natural wetlands that are relevant and meaningful for this research.  There are many indications (e.g., from Moreno-Mateos, D., et al. 2012. Structural and Functional Loss in Restored Wetland Ecosystems. PLoS Biology 10) that restored wetlands do not function ecologically in a manner equal to that of natural wetlands.  Chief among these differences are that soil nutrients, including N and P, and carbon are not equivalent in restored and reference wetlands.  Levels of these nutrients and of belowground organic matter in restored wetlands take decades to become equivalent to that found in reference wetlands.  The study reported here was conducted in wetlands that were restored around 15 years before the study was conducted.  Yet, the discussion and conclusions almost always infer that results apply to wetlands in general (although sometimes to urbanized wetlands).  I think the proper inference space in this study should be “restored wetlands” or perhaps “urbanized wetlands” but not “wetlands”.  Given the differences in nutrients and belowground organic carbon between wetlands of the age studied here and reference wetlands, surely microbial communities and even growth inhibition experiments using plants will have different outcomes.  I do not feel this consideration negates the findings or conclusions of the study, but proper inferences need to be drawn and a discussion of this issue of differences between restored and natural wetlands is important and should be added to the text.

One finding stands out to me; “complex mixtures of nutrients and metals in wetland ecosystems can produce relationships with bioindicators that run counter to the predictions made by considering individual pollutants independently”.  However, there isn’t much in the discussion to indicate why or what mechanisms might be involved to guide future research or any indication that such findings may be common. 

I stumble at bit with the style and content of the introduction.  It reads like an extended abstract to describe the study.  What are the strengths and weaknesses of the indices used here?  Why are plants and microbes used together (and not something else like macrofaunal communities and plants)?  Has this combination of metrics been used together in previous studies, or is it of interest because it is novel?  What has been found in previous studies that examine complex mixtures of nutrients and metals?  Finally, I don’t follow the intent of the second question (“can these bioindicators serve as predictive bioindicators of pollutant entering the wetlands”) nor do I see how it differs from the first question (also shouldn’t the word “pollutant” be “pollutants” in this second question?).  I’m not sure the authors even tried to address the second question in the discussion nor am I sure how it would be answered based on the results (perhaps I missed something?).

I’m a bit uncertain what to make of the contradictory results with different indicators (e.g., line 429, that states “The regression analyses for each plant bioindicator also revealed different relationships between inhibition and facilitation when analyzed for the effects of nutrient and metal pollution).  More study is suggested as a means to understand the differences between different indicator responses, but what does this do regarding the utility of these indicators in risk analysis?  Which indicators are better than others; what to do with contradictory results in risk analysis?  I think the authors should address these questions.

Minor comments:

Line 93, there are two periods after “August 2017”.

Line 265, should it read “in relation to phosphate concentrations (P<0.0001), and for nitrate…”?

Author Response

Cover Letter for Reviewer 1

Thank you for reviewing our paper. Please find below the answers to your questions. These edits have been incorporated in the paper as well. Please note that we encountered some problem with the "Track changes" and the "line numbers" working together. We have fixed the issue. But if some problem still exists please refer to the page and paragraph number that we have included below. I am attaching the edits along with this letter.

The questions comments and answers:

1. This research is an interesting approach to determine the potential ecosystem harm associated with complex mixtures of contaminants in freshwater wetlands.  Although no new specific methods were used to examine the combined effects of metals and nutrients, the combination of two disparate indicators (plant growth inhibition and the microbial community) appears to be novel and potentially valuable in ways that could lead to a better understanding of contaminant risk.  How best to examine complex mixtures is a difficult problem that requires new thinking; the approach used here may prove insightful.  However, like any new indicator, additional research will be needed before the meaning and full understanding of effects measured by these indices will be known.  This report raises many questions about the efficacy of the indicators used, especially given some contradictory results, and should stimulate additional research to refine the technique

Answer:

There are contradictions in the finding of the study, One example is, with the increase of Pb, an decrease in root inhibition was observed in the factor analysis. But a negative relationship in regression analysis was observed with the plant bioindicators and Pb concentration. I think these contradictions open up a great scope for further study. The future study can be in a lab controlled environment where the indicators can be tested against repeated stressors. Else, these indicators can be applied and be compared with natural wetlands. As the current study sites are “constructed wetlands” not natural and may not have attained their full ecological functional ability, so the comparison can bring up lot more explanation about the contradictions.

These contradictions have been separately addressed to acknowledge the scope of further study in the discussion section (See page 18, paragraph 4, lines 564-584).

Also the advantage of using two very different kind of bioindicators have been addressed in the discussion section. The bacterial one being quickly responsive and plants showing effect in the higher trophic level.(See page 19, paragraph 1, lines 586-598)

Some major comments:

2. The authors do not address differences between restored and natural wetlands that are relevant and meaningful for this research.  There are many indications (e.g., from Moreno-Mateos, D., et al. 2012. Structural and Functional Loss in Restored Wetland Ecosystems. PLoS Biology 10) that restored wetlands do not function ecologically in a manner equal to that of natural wetlands.  Chief among these differences are that soil nutrients, including N and P, and carbon are not equivalent in restored and reference wetlands.  Levels of these nutrients and of belowground organic matter in restored wetlands take decades to become equivalent to that found in reference wetlands.  The study reported here was conducted in wetlands that were restored around 15 years before the study was conducted.  Yet, the discussion and conclusions almost always infer that results apply to wetlands in general (although sometimes to urbanized wetlands).  I think the proper inference space in this study should be “restored wetlands” or perhaps “urbanized wetlands” but not “wetlands”.  Given the differences in nutrients and belowground organic carbon between wetlands of the age studied here and reference wetlands, surely microbial communities and even growth inhibition experiments using plants will have different outcomes.  I do not feel this consideration negates the findings or conclusions of the study, but proper inferences need to be drawn and a discussion of this issue of differences between restored and natural wetlands is important and should be added to the text.

Answer: These wetlands are not exactly restored wetlands. These are constructed freshwater wetlands built as a part of restoration project of Pike River, Racine WI, USA. These sites are used for stormwater retention and for ameliorating water quality before their water is discharged to the connected Pike River. But these are newly built wetlands hence the effect ecological functional maturity is still growing compared to natural wetlands. Hence we have mentioned specifically the age of the wetland sites, and then focused on that fact that these constructed freshwater wetlands are subject of investigation for ecological health using bioindicators, given the fact they are not very old (age varies between 13-20 years) and there are pollutant inputs like heavy metals and nutrients. This is especially required as these constructed freshwater wetlands not only function to retain storm water but also to ameliorating water quality (see page 1, paragraph 2, page 2 paragraph 1, lines 42-57).

3. One finding stands out to me; “complex mixtures of nutrients and metals in wetland ecosystems can produce relationships with bioindicators that run counter to the predictions made by considering individual pollutants independently”.  However, there isn’t much in the discussion to indicate why or what mechanisms might be involved to guide future research or any indication that such findings may be common. 

Answer: There are contradictions in the finding of the study, One example is, with the increase of Pb, an decrease in root inhibition was observed in the factor analysis. But a negative relationship in regression analysis was observed with the plant bioindicators and Pb concentration. I think these contradictions open up a great scope for further study. The future study can be in a lab controlled environment where the indicators can be tested against repeated stressors. Else, these indicators can be applied and be compared with natural wetlands. As the current study sites are “constructed wetlands” not natural and may not have attained their full ecological functional ability, so the comparison can bring up lot more explanation about the contradictions.

 This has been separately addressed to acknowledge the scope of further study in the discussion section. See page 18, paragraph 4, lines 564-584).

4. I stumble at bit with the style and content of the introduction.  It reads like an extended abstract to describe the study.  What are the strengths and weaknesses of the indices used here?  Why are plants and microbes used together (and not something else like macrofaunal communities and plants)?  Has this combination of metrics been used together in previous studies, or is it of interest because it is novel?  What has been found in previous studies that examine complex mixtures of nutrients and metals? 

Answer: We introduced the idea of using these specific indicators (ecotoxicological and bacterial) with reference to other studies and examples. Also have elaborated the justification of measuring these specific nutrients.(See page 2 to 3 and lines 63-105).  Such as, the bacterial indicator is quick in response to pollutants and plants shows effect in the higher trophic level. These combinations can reveal effect of pollutants in an ecosystem comprehensively from two tropic levels.

5. Finally, I don’t follow the intent of the second question (“can these bioindicators serve as predictive bioindicators of pollutant entering the wetlands”) nor do I see how it differs from the first question (also shouldn’t the word “pollutant” be “pollutants” in this second question?).  I’m not sure the authors even tried to address the second question in the discussion nor am I sure how it would be answered based on the results (perhaps I missed something?).

Answer: The goal of the paper is to identify two bioindicators that can correlate and predict response patterns in relation to specific pollutant such as nutrients (nitrate, phosphate) and heavy metals (such as Pb, Hg). Hence, keeping a second question as ”can these bioindicators serve as predictive bioindicators of pollutant entering the wetlands”, was a bit confusing. One question has been created instead of two:

“ Does variation in growth inhibition of plant  bioindicators and sediment bacterial taxonomical diversity correlate and predict response patterns with measured concentrations of nutrient and metal pollution (i.e. ex post impact indicators for ecological risk assessment) entering wetlands from the surrounding watershed?” (See page 3, paragraph 1, lines 100-104).

The discussion also now focuses on how these indicators are not only correlating with pollutants but also clearly demonstrating a specific response pattern (For example, relation with Pb and Hg is more clear compared to other detected metals). (See page 16-17, Discussion part 4.1 and 4.2.1)

6. I’m a bit uncertain what to make of the contradictory results with different indicators (e.g., line 429, that states “The regression analyses for each plant bioindicator also revealed different relationships between inhibition and facilitation when analyzed for the effects of nutrient and metal pollution).  More study is suggested as a means to understand the differences between different indicator responses, but what does this do regarding the utility of these indicators in risk analysis?  Which indicators are better than others; what to do with contradictory results in risk analysis?  I think the authors should address these questions.

Answer: There are contradictions in the finding of the study, One example is, with the increase of Pb, a decrease in root inhibition was observed in the factor analysis. But a negative relationship in regression analysis was observed with the plant bioindicators and Pb concentration. I think these contradictions open up a great scope for further study. The future study can be in a lab controlled environment where the indicators can be tested against repeated stressors. Else, these indicators can be applied and be compared with natural wetlands. As the current study sites are “constructed wetlands” not natural and may not have attained their full ecological functional ability, so the comparison can bring up lot more explanation about the contradictions.

 This has been separately addressed to acknowledge the scope of further study in the discussion section. See page 18, paragraph 4, line 564-584).

Minor comments:

Line 93, there are two periods after “August 2017”.

Answer: Dot removed. See page 3, paragraph 4, line 130

Line 265, should it read “in relation to phosphate concentrations (P<0.0001), and for nitrate…”?

Answer: See page 6, paragraph 6, line 304.

Reviewer 2 Report

The ms subject is interesting and is suitable for Diversity aim and scope. However, current version must be rewritten before acceptation.

First of all Authors have to read carefully an Instructions for Authors and prepare their ms according pointed rules.

References demands to be rewritten to the format acceptable by Diversity, the current version is completely unacceptable.

Line 16. Please precise the time of sampling, i.e. season of the year. Were the samples taken on the same dates each year?

In general Abstract section is not very informative, please pay an attention that methodologically right abstract should contain: aim, methodology and main conclusions. In the current version nothing is known about any methods and techniques you applied. Similarly, the reader not know about which pollutants you studied? In how many samples? etc.

Lines 31, 51. I suggest to add the following citation here: Banach A.M., Kuźniar, A., GrzÄ…dziel, J., WoliÅ„ska A., 2020. Azolla filiculoides L. as a source of metaltolerant microorganisms. PLoS ONE 15(5): e0232699. 

Lines 86-87. As sampling times was different (summer 2015 and 2017 and autumn 2016) the final results are difficult for interpretation and consequently conclusions could be not confirmed because time season of the year determine biodiversity changes, why you not sampled sediments during summer 2016?

Line 92. lack of space after [30]

Line 93. remove one dot

Lines 95, 108, 117. did you used any replicates of pointed measurements?

Line 130. did you isolated DNA one time from single sample?

Line 131. How long isolated DNA was stored?

Line 132. lack of space before [5]

Line 159, 275. in situ

Line 175. Why Nutrients, Metals and Factors are written with capital letters?

Line 200. lack of space before [29]

Lines 235-236. This fragment is suitable for methodology not for Results section

Line 248. orders

Line 264, 266: why Diversity is written with capital letter?

Line 280: “an increase in increase” – it is without any sense, please rewrite it

Citations of any literature position in Conclusions are improper, please rewrite this part so that it relates directly to the results obtained in carrying out the current work.

Author Response

Cover Letter for Reviewer 2

Thank you for reviewing our paper. Please find below the answers to your questions. These edits have been incorporated in the paper as well. Please note that we encountered some problem with the "Track changes" and the "line numbers" working together. We have fixed the issue. But if some problem still exists please refer to the page and paragraph number that we have included below. I am attaching the edits along with this letter.

The questions, comments and answers

The ms subject is interesting and is suitable for Diversity aim and scope. However, current version must be rewritten before acceptation.

First of all Authors have to read carefully an Instructions for Authors and prepare their ms according pointed rules.

1.References demands to be rewritten to the format acceptable by Diversity, the current version is completely unacceptable.

Answer: The references have been amended to be in the format required by Diversity for publication.(See page 21-24)

2.Line 16. Please precise the time of sampling, i.e. season of the year. Were the samples taken on the same dates each year?

In general Abstract section is not very informative, please pay an attention that methodologically right abstract should contain: aim, methodology and main conclusions. In the current version nothing is known about any methods and techniques you applied. Similarly, the reader not know about which pollutants you studied? In how many samples? etc.

Answer: The abstract has been rewritten. It is now sectioned into background information, aim, methods, results and significance. The pollutants and the sample timings also have been specified.

3.Lines 31, 51. I suggest to add the following citation here: Banach A.M., Kuźniar, A., GrzÄ…dziel, J., WoliÅ„ska A., 2020. Azolla filiculoides L. as a source of metaltolerant microorganisms. PLoS ONE 15(5): e0232699. 

Answer: Reference [11] added in page 2, paragraph 2 and line 58

4.Lines 86-87. As sampling times was different (summer 2015 and 2017 and autumn 2016) the final results are difficult for interpretation and consequently conclusions could be not confirmed because time season of the year determine biodiversity changes, why you not sampled sediments during summer 2016?:

Answer: The exact months of the sampling has been specified. (such as 2015: August,  2016: Late August and middle  September and 2017: August.) Field sampling was not feasible during 2016 summer due to lack assistance in the field. The days of sampling were close enough in time to not bring about issues of changes in biodiversity due to temperature or weather change. The temperature range in Wisconsin during August is 11-280C and in September is 15-230C (https://www.weather-us.com/en/wisconsin-usa-climate#climate_text_12)   (line: see 136-140)…See page 3, paragraph 3-4, line 122-134.

5.Line 92. lack of space after [30]. Space added. Reference number changed. See page 3, paragraph 5, line 129

6.Line 93. remove one dot : Extra dot removed. See page 3, paragraph 5, line 130

8.Line 130. did you isolated DNA one time from single sample?

Three extractions were done from every sample.(See page 4, paragraph 4, line 172-175)

9.Line 131. How long isolated DNA was stored?  DNA was stored for 72 hours.(See page 4, paragraph 4, line 175)

10.Line 132. lack of space before [5] space added, but reference number changed (See page 4, paragraph 4, line 178).

11.Line 159, 275. in situ Italics added (see page 4, paragraph 6, line 202)

12.Line 175. Why Nutrients, Metals and Factors are written with capital letters? Capital letters removed (see page 5, Section 3.1, line 218)

13.Line 200. lack of space before [29] space added See page 5, paragraph 6 line 243

14.Lines 235-236. This fragment is suitable for methodology not for Results section This fragment has been removed. See page 5, paragraph 3, line 281

15.Line 248. Orders plural form added. See page 6, paragraph 3, line 283.

16.Line 264, 266: why Diversity is written with capital letter?     Capital letters removed. See page 6, paragraph 6, line 303, 305

17.Line 280: “an increase in increase” – it is without any sense, please rewrite it “in increase” removed. See page 7, paragraph 2, line 318.

18.Citations of any literature position in Conclusions are improper, please rewrite this part so that it relates directly to the results obtained in carrying out the current work.

Answer: The conclusion has been rewritten without citations. See page 19, paragraph 3.

Round 2

Reviewer 1 Report

I feel the ms is not acceptable for publication

Reviewer 2 Report

Thank you for correction of the ms according my suggestions.

Author Response

Please see the attachment. We have incorporated extensive changes as suggested by the editor.
